# Veno-Venous Extracorporeal Membrane Oxygenation in COVID-19—Where Are We Now?

**DOI:** 10.3390/ijerph18031173

**Published:** 2021-01-28

**Authors:** Zbigniew Putowski, Anna Szczepańska, Marcelina Czok, Łukasz J. Krzych

**Affiliations:** 1Students’ Scientific Society, Department of Anaesthesiology and Intensive Care, Faculty of Medical Sciences in Katowice, Medical University of Silesia, 40-752 Katowice, Poland; mczok@poczta.fm; 2Department of Anaesthesiology and Intensive Care, Faculty of Medical Sciences in Katowice, Medical University of Silesia, 40-752 Katowice, Poland; a.j.szczepanska@gmail.com (A.S.); lkrzych@sum.edu.pl (Ł.J.K.)

**Keywords:** acute respiratory distress syndrome, acute respiratory failure, coronavirus disease 2019, veno-venous extracorporeal oxygenation

## Abstract

The recent development in extracorporeal life support (ECLS) has created new therapeutic opportunities for critically ill patients. An interest in extracorporeal membrane oxygenation (ECMO), the pinnacle of ECLS techniques, has recently increased, as for the last decade, we have observed improvements in the survival of patients suffering from severe acute respiratory distress syndrome (ARDS) while on ECMO. Although there is a paucity of conclusive data from clinical research regarding extracorporeal oxygenation in COVID-19 patients, the pathophysiology of the disease makes veno-venous ECMO a promising option.

## 1. Introduction

The recent development in extracorporeal life support (ECLS) has created new therapeutic opportunities for critically ill patients [1]. An interest in extracorporeal membrane oxygenation (ECMO), the pinnacle of ECLS techniques, has recently increased, as for the last decade, we have observed improvements in the survival of patients suffering from severe acute respiratory distress syndrome (ARDS) while on ECMO [2,3,4,5]. This technique enables an extracorporeal gas exchange, as blood flows through gas-permeable membranes and is exposed to fresh gas mixture, providing hemoglobin oxygenation and carbon dioxide removal.

Even though, ECMO in respiratory failure is not novel and the first attempts of its usage were introduced in 1971. For many years, its potential was neglected, mainly due to high mortality, severe adverse effects, limited accessibility, and high economic costs [6,7]. In adults, ECMO was primarily reserved for open-heart cardiac surgeries in which patients were cannulated to the aorta and inferior vena cava or the right atrium to take over the function of the heart and lungs (the so-called veno-arterial ECMO, VA-ECMO). Shortly afterwards, a modification to standard VA-ECMO was introduced, aimed to replace respiratory function only, namely, veno-venous ECMO (VV-ECMO). After several decades and after overcoming multiple technical obstacles, the role of VV-ECMO has been increasing over time, especially during outbursts of viral pneumonias [4]. This is all the more important given the research that has proven the harmful influence of invasive mechanical ventilation (MV) on the lung injury [8]. In patients with severe ARDS, parameters of mechanical ventilation necessary to maintain adequate oxygenation and carbon dioxide removal often exceed safe values, which paradoxically contributes to lung damage. In such patients, refractory to mechanical ventilation, VV-ECMO can provide optimal oxygen delivery while reducing ventilatory support (so-called ultraprotective mechanical ventilation) and thus, reduce ventilator-induced lung injury (VILI) [9,10]. It must be remembered, though, that VV-ECMO does not stand as a first-line treatment, is limited by several indications and contraindications, and constitutes a rescue therapy for patients who have a chance of recovery. The current COVID-19 pandemic has yet again increased the prevalence of ARDS, including the most severe cases, which naturally brings greater attention to the use of VV-ECMO, even as a bridge to lung transplantation for patients that suffered from critical damage inflicted over the course of the disease [11,12]. VA-ECMO is also a considerable option for those patients who additionally developed acute circulatory failure [13]. This review, however, is focused solely on VV-ECMO, as it is more commonly used and has a wider range of potential indications in patients infected with SARS-CoV-2.

## 2. A Few Words About the Procedure

In short, in order to perform VV-ECMO, double vascular access via a central vein is necessary. The cannulas should present certain properties, such as Fr of 23–31 for venous drainage and 15–19 Fr for venous return, in order to provide 3–7 L min^−1^ of blood flow [14]. Such a large size of the cannulas is a crucial factor in providing adequate blood flow, as there is a linear correlation between blood flow and oxygen transfer [15]. Usually, blood is drained from the femoral vein and is infused back in the internal jugular vein (Figure 1). Such an invasive vascular access is naturally associated with an increased risk of infections and therefore, special effort for the antiseptic care is required [14]. Apart from dual-vein access, double-lumen cannulas (DLCs) are also a viable option [16]. There are reports that the use of DLCs allows for a significant reduction of recirculation (i.e., repeated suction of already oxygenated blood by the inflow cannula) and more effective blood flow in the system [17].

The patient’s blood is moved from the vascular bed by a negative pressure generated by the pumps in the apparatus. The suction of blood can result in collapse of the veins, which disturbs extracorporeal blood flow and is more expressed in cases of hypovolemia, coughing, increased intra-abdominal pressure, and in malposition of cannulas [18]. 

Since blood passes through synthetic tubes, morphotic elements can be damaged and an activation of the inflammatory response and coagulation cascade occurs [19]. Thrombi can form all along an extracorporeal circuit, which not only disturbs the blood flow but can also lead to severe thrombotic events and the depletion of coagulation factors and blood platelets [20]. Hence, an anticoagulation is necessary. This is primarily achieved by systemic heparin-based anticoagulation; however, for patients with developed heparin-induced thrombocytopenia (HIT), a switch to alternative anticoagulation methods is necessary, namely, to direct thrombin inhibitors (e.g., bivalirudin or argatroban) [21]. Constant heparin infusions require regular monitoring of the activated partial thromboplastin time (APTT) or the activated clotting time (ACT). The elevated values of the latter parameters naturally pose a threat of systemic side effects (related to iatrogenic bleeding diathesis) [22]. However, recently, the idea of anticoagulation-free ECMO is emerging, as several studies reported a decreased incidence of bleeding events and no additional thrombosis risk in such a setting [23]. 

For the purpose of oxygenation and carbon dioxide removal, blood travels from the tubes and enters an oxygenator. In such a device, blood contacts with sweep gas via the polymethylpentene membrane through which the gas exchange occurs. The process of oxygenation depends on the blood flow and the concentration of O_2_ in fresh gas; therefore, for VV-ECMO, as stated before, high blood flows are necessary to keep an adequate arterial oxygen concentration. On the contrary, carbon dioxide removal depends only on the fresh gas flow, thus it can be performed with lower blood flows (so-called extracorporeal carbon dioxide removal—ECCO_2_R) [15,24]. Moreover, the entire process is dependent on the patient’s physiology as well; for VV-ECMO, heart function must be preserved and an adequate hemoglobin concentration is required for the oxygen transport to be sufficient [25]. Accordingly, if acceleration of blood flow does not optimize oxygenation, red blood cells transfusion can be considered as another way to improve oxygen delivery [26]. Nevertheless, as each blood product’s administration carries a risk of side effects, management of the hemoglobin level during VV-ECMO is still a matter of discussion [27]. 

In the first days of VV-ECMO or until a significant improvement in respiratory and function is achieved, deep sedation and muscle relaxation is recommended [28]. However, it should be remembered that sedation should be titrated to the light one in subsequent days and the pain–agitation–delirium algorithm should be implemented to provide effective physiotherapy and early mobilization [28,29]. Additionally, conducting the therapy in conscious patients (awake ECMO) is getting more common [30].

In summary, ECMO is associated with a number of severe adverse effects. Proper qualification for the procedure and compliance with existing guidelines are the first steps to avoiding those incidences. Frequent assessment of the cannula positioning, function of the oxygenator (blood gas analysis), and monitoring of coagulation parameters helps to reduce coagulopathic events (hemorrhage and thrombosis). Additionally, improvements in antiseptic behaviors reduce the rate of infections.

## 3. What Are the Indications and Contraindications for VV-ECMO?

The most important indication for VV-ECMO is any potentially reversible acute respiratory failure refractory to standard treatment, which has to be supervised firstly. The medical multidisciplinary team is obliged to optimize the patient’s condition as much as possible in a personalized manner (Table 1).

Most recognized indications and contraindications for VV-ECMO implementation are presented in Table 2. In the COVID-19 era, many of the mentioned indications became relative (see below).

Initiation of VV-ECMO in the times of the COVID-19 pandemic is restricted by other contraindications as well. As the entire procedure requires organization of medical personnel, equipment, and facilities, resources constraints force strict prioritization of patients that may benefit from the therapy [35]. VV-ECMO should not be commenced in patients in whom the therapy would be futile and should be reserved to subjects with better prognosis [32,36]. 

Thus, the qualification should be prudent and scientifically justified. Outcome predicting tools can be helpful in making decisions regarding initiation of ECMO. Currently, there is no ideal model that would confirm a positive outcome with diagnostic accuracy, mainly due to a plethora of specific complications of ECMO (e.g., bleeding and transfusion-related complications, thrombosis in the circuit, and organ-related complications). Therefore, the decision to undertake this therapeutic strategy should be based on pathophysiological rationale, clinical course, and the health burden due to comorbidities, taking into account the will of the patient. The factors most strongly associated with the risk of death during ECMO therapy, as determined before the initiation of therapy, are immunocompromised status, mechanical ventilation time, and SOFA score. Many mortality predictors using the above-mentioned factors are available; those most commonly applied for VV-ECMO are listed in Table 4.

Those scales were designed in the setting of typical ARDS. Many attempts have been made to create perfect prognostication models and their diagnostic accuracy has been confirmed to reach 80%. However, one ought to remember that in COVID-19, qualification to VV-ECMO is challenging and may be biased. The scales have not been validated in this specific clinical scenario yet. For instance, the PRESERVE score, with an AUROC of 0.89 (the highest one from those in Table 4), takes into account age, body mass index (BMI), immunological status, SOFA scale, duration of mechanical ventilation, prone positioning, PEEP, and plateau pressure values [38]. Unfortunately, the PRESERVE score lacks some of the key factors that are prominent in the severe COVID-19 population. For example, it is now known that high D-dimer levels independently correlate with the mortality of such patients [39,40]. Additionally, higher concentrations of lactate dehydrogenase (LDH) can early predict the severity of the disease [41]. Perhaps considering those markers when assessing risk of death in acute respiratory failure due to COVID-19 would be helpful in distinguishing patients with the best chance of survival and benefiting from VV-ECMO. Moreover, plateau pressure and PEEP values may not fully reflect the severity of COVID-19-derived acute respiratory failure (ARF), as many patients with this disease, especially at the early stages, present high-compliance lungs but still suffer from refractory hypoxemia [42]. Therefore, such patients will not achieve plateau pressures above 30 cmH_2_O, but their prognosis may still be doubtful. Additionally, prone positioning in such patients may not be recommended [42]. Lastly, the PRESERVE score shows that BMI above 30 kg/m^2^ serves as a factor that improves survival; however, in COVID-19, obesity is strongly associated with the mortality and with severity of the disease [43]. The above-mentioned issues indicate a necessity for updated outcome predicting tools when selecting COVID-19 patients for the VV-ECMO therapy.

## 4. VV-ECMO Efficacy: Glimpse of the Recent Past

Scientific data from randomized trials regarding VV-ECMO in respiratory failure is scarce. In 2009, a multicenter trial for the efficacy and economic assessment of Conventional ventilatory support versus Extracorporeal membrane oxygenation for Severe Adult Respiratory failure (CESAR) was performed [3]. In this study, 180 patients were enrolled, 90 of which were assigned for the VV-ECMO implementation. The eligibility criteria included patients with severe but potentially reversible ARF, a Murray score of 3 or higher, and pH < 7.20 despite optimal ventilatory management. For the intervention group, the median time for ECMO initiation from the time of randomization was 6.1 h (40% of ECMO patients were already ventilated for longer than 48 h). For the ventilatory support, no strict criteria were introduced, as clinicians were only advised to maintain protective mechanical ventilation (low tidal volume and plateau pressure below 30 cmH_2_O). Notably, 25% of all patients randomized to the intervention group did not receive ECMO (importantly, primary analysis was by intention to treat). The researchers found the mortality (or severe disability) in patients with ECMO was 37%, compared to the conventional group, where it was 53% (RR 0.69; 95% CI 0.05–0.97, *p* = 0.03). It is possible that due to the extracorporeal gas exchange, less invasive ventilatory parameters could be attained (ultraprotective mechanical ventilation), therefore reducing the prevalence of ventilatory-induced lung injury (VILI) and mortality. One of the limitations of the above-mentioned study is that mechanical ventilation treatment was not standardized, and this could have resulted in higher pressures being implemented in the conventional group, therefore promoting VILI.

One of the most frequently discussed studies that yet again made ECMO controversial was the Extracorporeal Membrane Oxygenation for Severe Acute Respiratory Distress Syndrome (EOLIA) trial [2]. The EOLIA was a multi-center randomized controlled trial in which patients with severe ARDS were randomized to either early implementation (median time = 3.3 h) of VV-ECMO or conventional mechanical ventilation with a possibility to start ECMO as a rescue therapy in case of refractory hypoxemia (SaO_2_ < 80% for >6 h). The study protocol provided detailed criteria for mechanical ventilation. In the study group, the following ventilation settings were implemented: volume-assist control mode, FiO_2_ = 30–50%, PEEP > 10, plateau pressure < 24 cmH_2_O, and respiratory rate 10–30 breaths per minute; while in the control group, ventilation with TV maximum of 6 mL/kg of ideal body weight and plateau pressure < 30 cmH_2_O was applied. In addition, physicians were encouraged to apply adjunctive interventions, such as prone positioning, the use of neuromuscular blocking agents, inhalation of nitric oxide, recruitment maneuvers, high-frequency oscillatory ventilation, or almitrine infusion in the control group, if necessary. The study was terminated prematurely due to no significant differences in the 60-day mortality (primary endpoint) between the groups in the interim analysis of data obtained after enrollment of 240 of the pre-specified 331 patients optimal for statistical analysis (35% in the ECMO group, 46% in the control group; RR 0.76; 95% CI 0.55–1.04; *p* = 0.09). Unfortunately, early cessation made the entire study statistically underpowered and unable to detect significant differences in mortality. Finally, 249 patients were randomized: 124 to the early ECMO group and 125 to the control group; however, as many as 35 (28%) patients in the control group required conversion to ECMO, which could have falsely reduced the difference in mortality between the groups. If the crossover was not allowed by the study protocol, the survival rate in the control group would have been much lower (15 out of 35 patients placed on the rescue ECMO survived) and thus the relative risk reduction would have increased to a significant value. Moreover, the risk of treatment failure (defined as death by day 60 in patients in the interventional group and as crossover to ECMO or death in patients in the control group) was significantly lower in the early ECMO group in comparison to the control group (RR 0.62, 95% CI 0.47–0.82; *p* < 0.05). Furthermore, post-hoc Bayesian analysis pointed out a very high probability of improving outcome in severe ARDS if ECMO was applied (88% to 99% chance of success depending on the chosen priors) [44].

Recently, EOLIA and CESAR data were put into a meta-analysis and the investigators concluded that indeed, there was evidence that VV-ECMO reduced mortality and organ failure [45]. Importantly, this study showed that the implementation of ECMO in patients with one or two organ failures significantly reduced mortality (22% vs. 41%) compared to patients with three or more organ failures. This suggests that ECMO may improve the outcome only in patients in whom multi-organ failure and severe shock has not yet occurred. Those with multiorgan failure would not benefit from this invasive technique. In conclusion, early implementation of VV-ECMO in severe ARDS may significantly improve the outcome under specific conditions only. 

## 5. Is There a Place for VV-ECMO in the Times of the COVID-19 Pandemic?

Based on the above-mentioned information, VV-ECMO may represent a valuable therapeutic option in patients with severe lung failure in the course of the SARS-CoV-2 infection. Importantly, there are recommendations that very specifically define the indications for ECMO therapy in COVID-19 in several countries [46]. It is important that these recommendations relate to the period of the pandemic when ICU admission, discharge, and triage guidelines should be adjusted to local needs and abilities. Nontraditional indications and contraindications are usually necessary in such a situation. Thus, they allow the age criterion to be changed depending on the course of the pandemic. There is a limited number of observations regarding VV-ECMO in COVID-19 patients (Table 5). Individual data suggest that early implementation of VV-ECMO may improve the outcome [2,45]. 

One ought to know that the mortality in patients treated with ECMO in the above-described studies (i.e., Schmidt and Barbaro) was comparable with EOLIA (35%) and CESAR (37%). In the LIFEGUARDS multicenter prospective observational study, mortality reached 39% [55]. This similarity illustrates the potential effect in mortality reduction that can be expected in COVID-19 patients with ARF unresponsive to standard ventilatory treatment. 

COVID-19-related ARF is not only the pure ARDS due to viral pneumonia, so the prognostication about the course of disease may be extremely difficult. The current understanding of severe COVID-19 indicates that at the roots of ARF, due to the infection, is the so-called endothelialitis, that is, the inflammation of the endothelium, which occurs mainly in the lung vasculature [56]. SARS-CoV-2 infection often leads to a procoagulable state, which results in microthrombi (or immunothrombi) formation, mainly in the peripheral pulmonary arteries [57]. Italian researchers propose to use the term “MicroCLOTS” (microvascular COVID-19 lung vessels obstructive thromboinflammatory syndrome) as a new name for severe pulmonary COVID-19 [58]. They suggest that the activation of the complement cascade leads to endothelial damage and recruits leucocytes. This reaction is responsible for a massive local release of proinflammatory cytokines, such as interleukin (IL)-1, IL-6, IL-8, and Interferon gamma (IFN-γ). Of note, such a robust inflammatory response probably makes COVID-19 patients biologically predisposed to ventilator-induced lung injury (VILI) and, therefore, the risk of VILI should be minimized by introducing lung protective ventilation [58,59]. Unfortunately, maintaining protective ventilatory parameters in those patients may result in increasing hypoxia and hypercapnia, mainly due to low tidal volumes. This makes the ECLS technique a viable option in either reducing the power of mechanical ventilation or improving gas exchange. Nevertheless, due to the host immune response, many immune cells, namely lymphocytes, macrophages, monocytes, and neutrophils, run their proinflammatory functions, leading to microvascular thrombosis vascular endothelial and alveolar cell damage [58]. Additionally, such a hypercoagulable state might be augmented by the increase of pathological antibodies, often of similar characteristics to those found in antiphospholipid syndrome [60]. There are some laboratory changes that are associated with thrombotic complications in COVID-19 (Table 6). VV-ECMO with therapeutic anticoagulation may therefore help to reduce the risk of clinical consequences of hypercoagulability. 

In the most severe COVID-19 patients, ARF may, therefore, be a result of impaired vascular function and hemostatic balance, and not lung tissue itself. Indeed, many patients with early COVID-19-derived respiratory failure present with high-compliance lungs, which means that an increase in PEEP values may not improve oxygenation and can even worsen it [62,63]. High mean airway pressure, especially in high-compliance lungs, can increase pulmonary vascular resistance even to a point of blood flow obstruction. This results in higher ventilation-to-perfusion ratios and, therefore, in expansion of West zone 1 (where the pressure in airways surpasses pressures in the pulmonary arteries and veins). Such a condition is considered as an increased dead space and worsens gas exchange. It must also be remembered that dead space results in hypercapnia, which has a number of detrimental effects, including inhibition of alveolar wound repair, proliferation of alveolar cells, and reabsorption of alveolar fluid [59,64]. In this context, patients with high pulmonary compliance may be refractory to the standard ventilatory treatment and other supportive procedures, such as prone positioning. Implementation of VV-ECMO in these conditions could bypass the above-mentioned issues and could additionally reduce hypoxic pulmonary vasoconstriction and right ventricle afterload, as upon entering pulmonary vasculature, blood would already be well saturated with the oxygen [7,63]. 

As it is clear now, COVID-19-related ARF is a highly complex pathology, where taking into account only the clinical picture of lungs is scientifically insufficient. Therefore, based on the quartiles of D-dimer concentrations and lung static compliance, Grasselli et al. divided COVID-19 patients with ARF into four groups [65]. The first group, namely, high D-dimers and low lung compliance (HDLC) with D-dimer concentrations greater than the median in COVID-19 ARDS (>1880 ng/mL) and static compliance equal to or less than the median (41 mL/cm H_2_O), constituted 27% of COVID-19 ARDS patients. The low D-dimers and high compliance (LDHC) group contained patients exhibited D-dimer concentrations equal to or less than the median, and static compliance greater than the median (26% of all patients). The low D-dimers and low compliance (LDLC) group included patients with D-dimer concentrations and static compliance equal to or less than the medians (23% of all patients). The high D-dimers, high compliance (HDHC) group was patients with D-dimer concentrations and static compliance greater than the medians (24% of all patients). Interestingly, the HDLC group had significantly higher 28-day mortality than the other three groups (56% in the HDLC group vs. 27% in the LDHC group, 22% in the LDLC group, and 35% in the HDHC group, all *p* = 0·0001).

As it is seen above, the severity of the disease depends both on the lungs condition and coagulation status. Indeed, the administration of anticoagulants improves survival [66]. Patients receiving ECMO would therefore benefit not only from improved gas exchange but also from receiving large doses of anticoagulation drugs. However, anticoagulation in those patients is often insufficient as it is reported that the majority of patients receiving ECMO suffer from thrombotic occlusions within centrifugal pumps [67]. An additional therapeutic option for patients with SARS-CoV-2 infection requiring ECMO is the adhibition of a procoaguable cytokine elimination filter, which is thought to attenuate the cytokine storm, including IL-6, which is regarded as a procoaguable cytokine [68]. There are ongoing studies in this area, which may answer the question whether the reduction of cytokine levels during ECMO therapy improves survival in this group of patients [69]. 

## 6. Conclusions

VV-ECMO is an effective viable therapeutic option for severe acute respiratory failure, which is refractory for conventional treatment. Although there is a paucity of conclusive data from clinical research regarding extracorporeal oxygenation in COVID-19 patients, the pathophysiology of the disease makes VV-ECMO a promising option. 

## Figures and Tables

**Figure 1 ijerph-18-01173-f001:**
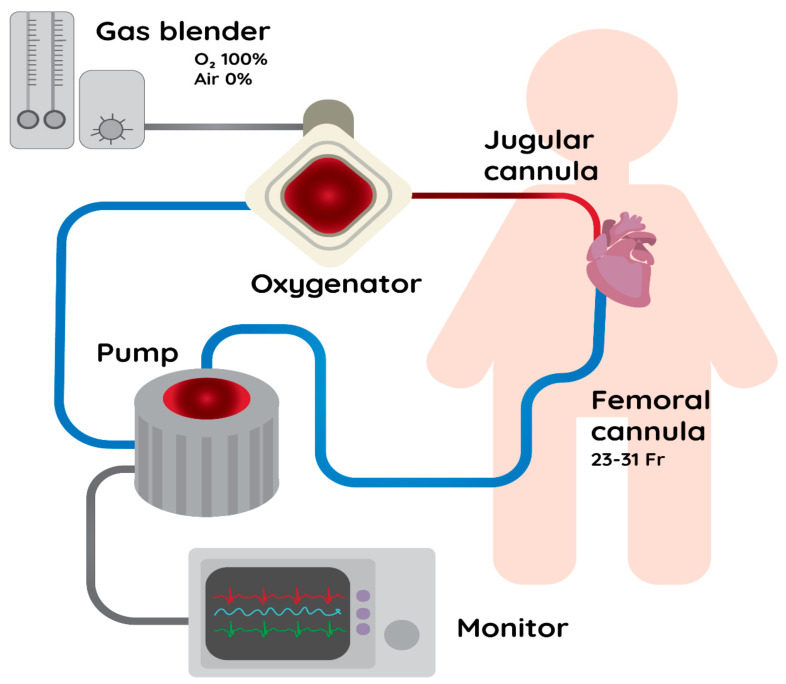
A scheme of the VV-ECMO circuit. VV-ECMO– Venovenous Extracorporeal Membrane Oxygenation.

**Figure 2 ijerph-18-01173-f002:**
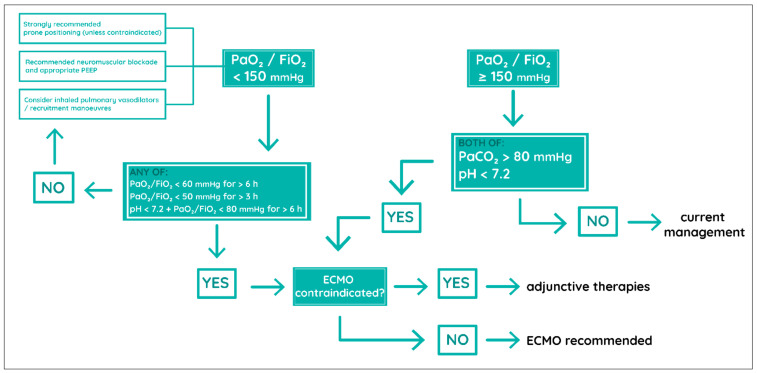
Conventional VV-ECMO indications for ARDS ([32], self-modified). VV-ECMO—Venovenous Extracorporeal Membrane Oxygenation, ARDS—Acute Respiratory Distress Syndrome, PaO_2_—partial pressure of arterial oxygen, FiO_2_—fraction of inspired oxygen, PaCO_2_—partial pressure of carbon dioxide.

**Table 1 ijerph-18-01173-t001:** Optimization procedures before VV-ECMO qualification [31].

Exclusion or removal of potentially reversible causes of deterioration in lungs function: pneumothorax, significant pleural effusion, bronchial obstruction with respiratory secretion or clot, congestion in pulmonary vasculature, increased extravascular lung water
Protective mechanical ventilation: VT ≤ 6 mL kg^−1^ of ideal body weight according to the ARDSNet table, Pplat < 30 cmH_2_O, permissive hypercapnia, driving pressure < 14 cmH_2_O
Adequate sedation (RASS -4/-5). If there is poor tolerance of ventilation with low tidal volumes and difficulties in patient-ventilator synchronization, in severe cases of ARDS (PaO_2_/FiO_2_ < 120 mmHg), implementation of muscle relaxants for a maximum of 48 h should be considered
PEEP titration (5-10-15-20 cmH_2_O) to optimal lung compliance considering hemodynamic effects and optimal PaO_2_/FiO_2_ values, preferably by derecruitment technique
Frequent bronchial tree toilet (closed suction system), daily bronchoscopy + subsequent recruitment maneuvers
Optimization of fluid therapy—negative fluid balance (forced diuresis, CRRT), preferably according to EVLW (extra-vascular lung water) < 10 mL kg^−1^
Optimization of the circulatory system and appropriate vasopressor support
If prone positioning results in significant improvement of oxygenation, it should be implemented at least twice a day for 6–8 h + recruitment maneuvers in the prone position and determination of optimal PEEP
To reduce the risk of pneumonia associated with mechanical ventilation: rational antibiotic therapy, avoiding reintubation, gastric probe inserted through the mouth, elevation of the head of the bed to 30°–45°, suction from above the cuff of the endotracheal tube, monitoring depth of sedation, early enteral nutrition, glycemia control, peptic ulcer prophylaxis (sucralfate), venous thromboembolism prophylaxis

Pplat—plateau pressure, RASS—Richmond Agitation-Sedation Scale, ARDS—Acute Respiratory Distress Syndrome, PaO_2_- Partial pressure of arterial oxygen, FiO_2_—Fraction of inspired oxygen, CRRT—Continuous Renal Replacement Therapy, EVLW—Extravascular Lung Water.

**Table 2 ijerph-18-01173-t002:** Indications and contraindications for ECMO ([32,33], self-modified).

Indications	Contraindications
Potentially reversible acute respiratory failure	Irreversible cardiac or pulmonary disease
Severe ARDS refractory to standard treatment (Figure 2)	Severe brain injury
Pulmonary thromboembolism with preserved cardiac function	Polytrauma with a high risk of bleeding
Trauma (pulmonary contusion)	Severe pulmonary hypertension
Murray score ≥ 3 pts (Table 3)	Uncontrolled bleeding
Failed lung transplant graft	Mechanical ventilation for >14 days before initiation of ECMO or ventilation at high settings (FiO_2_ > 0.9, Pplat > 30) for ≥7 days

ARDS—Acute Respiratory Distress Syndrome, ECMO—Extracorporeal Membrane Oxygenation, FiO_2_—Fraction of inspired oxygen, Pplat—Plateau Pressure.

**Table 3 ijerph-18-01173-t003:** Murray score for the initiation of VV-ECMO in acute respiratory failure [34].

Parameter/Score	0	1	2	3	4
PaO_2_/FiO_2_ (mmHg)	≥300	225–299	175–224	100–174	<100
Chest X-Ray (quadrants infiltrated)	normal	1	2	3	4
PEEP (cmH_2_O)	≤5	6–8	9–11	12–14	≥15
Compliance (ml/cmH_2_O)	≥80	60–79	40–59	20–39	≤19

PaO_2_—Partial pressure of arterial oxygen, FiO_2_– Fraction of inspired oxygen, PEEP—Positive End-Expiratory Pressure.

**Table 4 ijerph-18-01173-t004:** Mortality prediction scales for patients who may undergo VV-ECMO [37].

Name	Mode	AUROC	95% CI
RESP score	VV. VA	0.74	0.72–0.76
ECMOnet score	VV	0.86	0.74–0.96
Score by Roch et al.	VV	0.8	0.71–0.89
PRESERVE score	VV	0.89	0.83–0.94
Score by Enger et al.	VV	0.75	NA
VV ECMO mortality score	VV	0.76	0.67–0.85

AUROC—Area Under the Receiver Operating Characteristics, VV—venovenous, VA—venoarterial, VV ECMO—Venovenous Extracorporeal Membrane Oxygenation, CI—confidence interval

**Table 5 ijerph-18-01173-t005:** A summary of trials in which VV-ECMO was implemented in severe acute respiratory failure associated with COVID-19 (as of 15 November 2020). MV—Mechanical ventilation.

Study	Type of the Study	Population	Time of VV-ECMO Implementation	Effect
Liu et al. [47]	Retrospective observational	6 patients	12 days (median time) after MV initiation	No patient who received ECMO died (at day 28 from admission)
Osho et al. [48]	Prospective observational	6 patients	5.5 days (median time) after MV initiation	1 patient died during ECMO
Beyls et al. [49]	Retrospective observational	12 patients	4 days (median time) after MV initiation	2 patients died during ECMO (however, at the time of analysis, 8 patients were still on ECMO)
Schmidt et al. [50]	Retrospective observational	83 patients	4 days (median time) after MV initiation	30 patients (36%) died. Importantly non-survivors received ECMO later than survivors (6 days vs. 4 days)
Falcoz et al. [51]	Prospective observational	16 patients	4 days (median time) after MV initiation	6 patients died (mortality at day 60 was 38%)
Sultan et al. [52]	Retrospective observational	10 patients	3 days (median time) after MV initiation	1 patient died (mortality was assessed on day 9)
Mustafa et al. [53]	Retrospective observational	40 patients	4 days (mean time) after MV initiation	6 patients died (15%). 29 patients (73%) have been discharged from the hospital.
Barbaro et al. [54]	Retrospective, observational	1035 patients	4 days (median time) after endotracheal intubation	38% mortality at day 90

**Table 6 ijerph-18-01173-t006:** Laboratory changes often found in severe COVID-19 infection ([61], self-modified).

D-Dimer	Elevated
Fibrinogen	Elevated
FDPs	Elevated
Platelet count	Normal or mildly decreased
Plasma viscosity	Increased
Factor VIII activity	Increased
Von Willebrand factor	Increased
Protein C i S	Modestly decreased

FPDs—Fibrinogen degradation products.

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
