# Peer review of "Veno-Venous Extracorporeal Membrane Oxygenation in COVID-19—Where Are We Now?"

_ijerph, 2021, doi:10.3390/ijerph18031173_

Round 1

Reviewer 1 Report

the topic of this research, the extracorporeal life support like extracorporeal membrane oxygenation ECMO, is important in pandemia, mostly a modification to standard VA-ECMO is fundamental, and was introduced, aimed to replace respiratory function called veno-venous ECMO (VV-ECMO).  Covid-19-related acute respiratory failure  is a highly complex pathology of lungs, in the most severe Covid-19 patients acute respiratory failure may, therefore, be a result of impaired vascular function and hemostatic balance, but not lung tissue. in this manuscript it was proved  that V-ECMO is an effective, viable therapeutic option for severe acute respiratory failure, which is indeed refractory for conventional treatment. As it is shown, some conclusive data from clinical research regarding extracorporeal oxygenation in Covid-19 patients, pathology and physiology of the disease makes VV-ECMO a promising option. paper is sound, with tables and Good figures depicting the problema.please check carefully english language.

Thank you very much for your appreciative review.

We have checked English writing.

Author Response

Thank you very much for your appreciative review.

We have checked English writing.

Reviewer 2 Report

Dear Editors,

Thank you for the opportunity to review this manuscript. This manuscript has the goal of reviewing VVECMO particularly in the setting of the COVID-19 setting.  Moreover, they go over briefly VVECMO as a procedure.  I believe this is very well written and comprehensive. They go over the pertinent literature regarding COVID19 and ECMO.  

What is extra impressive is the amount of effort the authors go into the indications of VVECMO.  In particular, they highlight the need for the indication to be a correctable insult which is of grave importance during a pandemic where resources are highly limited.  We would recommend to accept for publication

There are minimal changes that we would like to have in the revisions including the following:

Thank you for your appreciative review.

1). The proper term for the pandemic is COVID-19 not Covid-19. Please change accordingly

1) We have corrected it.

2).Is there any use in commenting on t he use of VA-ECMO in this setting?

2) We have added a sentence that implicates possible use of VA ECMO in COVID-19 population.

3). What is extra impressive is the amount of effort the authors go into the indications of VVECMO.  In particular, they highlight the need for the indication to be a correctable insult which is of grave importance during a pandemic where resources are highly limited. Can you offer some more guidance regarding this portion.  In the setting of COVID-19, what is "correctable"?

3) All acute respiratory failures are potentially reversible and the same applies for COVID-19-derived acute respiratory failure. The key to assess whether a patient can benefit from VV-ECMO is to carefully consider patient's comorbidities, prognosis and overall general state (provided by indications and contraindications). One cannot foresee the potential "reversibility" of COVID-19 ARF, therefore, it has to be assumed. We believe we described this issue in details in our manuscript.

4). Patients are getting transplants for COVID infection.  Some of these patients will likely require ECMO as bridge to transplantation.  There should be some mention of this in the manuscript as this is an application that is already well known in the literature

4) We have added a sentence regarding this matter in the Introduction.

Thank you for this contribution to the literature.

Author Response

Thank you for your appreciative review.

1) We have corrected it.

2) We have added a sentence that implicates possible use of VA ECMO in COVID-19 population.

3) All acute respiratory failures are potentially reversible and the same applies for COVID-19-derived acute respiratory failure. The key to assess whether a patient can benefit from VV-ECMO is to carefully consider patient's comorbidities, prognosis and overall general state (provided by indications and contraindications). One cannot foresee the potential "reversibility" of COVID-19 ARF, therefore, it has to be assumed. We believe we described this issue in details in our manuscript.

4) We have added a sentence regarding this matter in the Introduction.

Reviewer 3 Report

The authors present overall review of VV-ECMO especially during Covid-19 era. It is well written and summarized in terms of the indication, contraindication, effectiveness of VV-ECMO based on proper references. The study is overall well-written, but I have some specific comments.

Thank you for your appreciative review.

  • Please describe the difference between conventional catheter system and bi-caval system in terms of re-circulation ratio and others. You write CESAR study (2006) and EOLIA study (2018). Different systems may be used in each study.

1) We described the differences between single-lumen and double-lumen cannulas. In EOLIA and CESAR study, the properties of catherers are not given, therefore, we did not mention it.

  • Line 110: two periods are described.

2) We corrected it.

  • Table 2. As you mentioned, the age of contraindication for ECMO is not determined. This review is for patients with COV-19; therefore, you had better delete “Age > 65 Year”.

3) We deleted it.

  • Line 235, Table 5: Please show the abbreviation of MV.

4) We added abbreviation.

Author Response

Thank you for your appreciative review.

1) We described the differences between single-lumen and double-lumen cannulas. In EOLIA and CESAR study, the properties of catherers are not given, therefore, we did not mention it.

2) We corrected it.

3) We deleted it.

4) We added abbreviation.

Reviewer 4 Report

The manuscript entitled “Veno-venous extracorporeal membrane oxygenation in Covid-19 - where are we now?” can be accepted by ijerph after major revision. Overall, novel Veno-venous extracorporeal membrane oxygenation is a multi center prospective observational study that has been used to explore the role of ECMO in the treatment of COVID-19. The results of this study provide a practical basis for the application of Veno-venous extracorporeal membrane oxygenation.

Thank you for your review.

1. This is a review manuscript, and there is a lack of experimental data, so I think the authors should add the introduction of related work to show that the authors' hypothesis has a solid theoretical basis.

1) It is not possible to provide adequate experimental data, as majority of research done on VV-ECMO in COVID-19 is solely on clinical human data. Our hypotheses are, therefore, based on clinical implications and already existing clinical data. For instance, please see references 42,55,56,58,66.

2. Extracorporeal membrane oxygenation is risky. It draws the patient's venous blood out, passes through an oxygenator, and then goes back to another vein through the machine. This can easily cause bleeding, blood clotting and possibly infection. The author should add some comments and prospects to solve these potential problems.

2) We added a summary regarding correct monitoring of ECMO to reduce the number of adverse effects.

Author Response

Thank you for your review.

1) It is not possible to provide adequate experimental data, as majority of research done on VV-ECMO in COVID-19 is solely on clinical human data. Our hypotheses are, therefore, based on clinical implications and already existing clinical data. For instance, please see references 42,55,56,58,66.

2) We added a summary regarding correct monitoring of ECMO to reduce the number of adverse effects.

Round 2

Reviewer 4 Report

The author has optimized the manuscript better, so I think it can be accepted.